# Impacts of Telomeric Length, Chronic Hypoxia, Senescence, and Senescence-Associated Secretory Phenotype on the Development of Thoracic Aortic Aneurysm

**DOI:** 10.3390/ijms232415498

**Published:** 2022-12-07

**Authors:** Thomas Aschacher, Daniela Geisler, Verena Lenz, Olivia Aschacher, Bernhard Winkler, Anne-Kristin Schaefer, Andreas Mitterbauer, Brigitte Wolf, Florian K. Enzmann, Barbara Messner, Günther Laufer, Marek P. Ehrlich, Martin Grabenwöger, Michael Bergmann

**Affiliations:** 1Department of Cardiovascular Surgery, Clinic Floridsdorf and Karl Landsteiner Institute for Cardio-Vascular Research, 1210 Vienna, Austria; 2Department of Cardiac Surgery, Medical University of Vienna, 1090 Vienna, Austria; 3Department of Plastic, Reconstructive and Aesthetic Surgery, Medical University of Vienna, 1090 Vienna, Austria; 4Faculty of Medicine, Sigmund Freud Private University of Vienna, 1030 Vienna, Austria; 5Department of General Surgery, Medical University of Vienna, 1090 Vienna, Austria; 6Department of Vascular Surgery, Medical University of Innsbruck, 6020 Innsbruck, Austria

**Keywords:** telomere, aneurysm, DNA damage, senescence-associated secretory phenotype, cell death, thoracic aorta, telocytes

## Abstract

Thoracic aortic aneurysm (TAA) is an age-related and life-threatening vascular disease. Telomere shortening is a predictor of age-related diseases, and its progression is associated with premature vascular disease. The aim of the present work was to investigate the impacts of chronic hypoxia and telomeric DNA damage on cellular homeostasis and vascular degeneration of TAA. We analyzed healthy and aortic aneurysm specimens (215 samples) for telomere length (TL), chronic DNA damage, and resulting changes in cellular homeostasis, focusing on senescence and apoptosis. Compared with healthy thoracic aorta (HTA), patients with tricuspid aortic valve (TAV) showed telomere shortening with increasing TAA size, in contrast to genetically predisposed bicuspid aortic valve (BAV). In addition, TL was associated with chronic hypoxia and telomeric DNA damage and with the induction of senescence-associated secretory phenotype (SASP). TAA-TAV specimens showed a significant difference in SASP-marker expression of IL-6, NF-κB, mTOR, and cell-cycle regulators (γH2AX, Rb, p53, p21), compared to HTA and TAA-BAV. Furthermore, we observed an increase in CD163^+^ macrophages and a correlation between hypoxic DNA damage and the number of aortic telocytes. We conclude that chronic hypoxia is associated with telomeric DNA damage and the induction of SASP in a diseased aortic wall, promising a new therapeutic target.

## 1. Introduction

Aortic aneurysms are characterized by local inflammation with degeneration around the aorta, leading to vessel weakening and dilatation [1]. Degenerative remodeling in the medial layer of aortic aneurysm tissue is characterized by loss of vascular smooth muscle cells (vSMC) and destruction of the extracellular matrix (ECM) [2,3]. This medial degeneration leads to weakening and progressive dilatation of the vascular wall, and ultimately results in aortic dissection or aneurysm rupture. In addition to luminal blood flow, the aortic wall also depends on the vasa vasorum, a microvascular network that nourishes the adventitia and smooth muscle cells in the medial layer [4]. Apoptosis of vSMC is characteristic for aortic aneurysms, and Billaud et al. showed that not only chronic hypoxia in the medial layer but also adventitial remodeling of the vasa vasorum with down-regulation of angiogenic signaling (metallothionein 1A and the pro-angiogenic factor vascular endothelial growth factor) are essential in the development of aneurysms [5]. A chronic hypoxic environment of vSMC, epidermal cells, endothelial cells (ECs) and blood lymphocytes leads to the occurrence of oxidative damage in telomeric DNA, which is significantly higher in the wall of abdominal aortic aneurysms [6]. Telomeres of vSMC, epidermal cells, ECs, and blood lymphocytes are significantly shortened in these patients [6,7].

Telomeres are repetitive DNA sequences at the ends of chromosomes that maintain genomic stability. After each cell division, telomeres shorten, and when reaching a critical state, they trigger a DNA damage response (DDR) or senescence [8,9]. Six Shelterin proteins protect the structure of telomeres [10]. If they fail to bind to telomeric DNA, telomeres are uncapped and may trigger senescence, apoptosis, end-to-end fusions, and genomic instability [8,9,11,12]. The histone protein 2AX (H2AX) is a central component of numerous signaling pathways in response to double-stranded breaks (DSB), which becomes rapidly phosphorylated to form γH2AX at nascent DSB sites [13]. When truncated telomeres fail to protect the end of chromosomes, the uncovered double-stranded end induces formation of γH2AX foci, which are excellent markers of telomere erosion, and hence, replicative senescence [14]. Senescence is defined as the irreversible cessation of cell division [15]. Once the process of senescence is activated, cellular changes occur which lead the cell to a senescence-associated secretory phenotype (SASP) [16,17]. Cells with SASP release several characteristic proteins, including growth factors, chemokines, cytokines, and metalloproteases [18,19,20]. The classic signaling pathway, triggered in response to DNA damage, involves the activation of ATM and ATR kinases, which activate the p53/p21 protein [21]. While p53 and p21 are essential for the initiation of senescence, ATM plays an important role in inducing the SASP phenotype [21].

The permanent senescence of many SASP cells contributes to faster development of age-related diseases, including diabetes and atherosclerosis [16,22,23]. In contrast to induced senescence, which is triggered by shortened telomeres, senescence of postmitotic cells does not depend on the length of telomeres but on their functionality [16]. Some proteins of the Shelterin complex also play a protective role by inhibiting senescence-initiating kinases and limiting the production of reactive oxygen species (ROS) by mitochondria. During chronic senescence, morphological changes occur. Cells become large and flatten, their outline becomes more irregular, and the number of vacuoles increases. Senescent cells show accumulation of dysfunctional mitochondria and increased release of ROS [24].

While numerous diseases, adverse lifestyle habits, and environmental factors accelerate telomere shortening, oxidative stress is the most common biological cause [25,26,27,28]. Oxidative stress induces an increased concentration of ROS. ROS, a source of the inflammatory hypoxic environment generated by immune cells, induce single-strand breaks in telomeric DNA and lead to telomere loss [27,29]. The altered homeostasis of telomere length plays a crucial role in the aortic aneurysm etiology, a disease with life-threatening complications [26,30]. In a murine model, ROS are detected in aortic aneurysms, whereas knockout mice with suppressed ROS production have reduced aneurysmal formation [31,32].

The aim of this study was to investigate whether telomeres are also shortened in thoracic aortic aneurysms (TAA). We also compared telomeric DNA damage in patients with normal tricuspid aortic valves (TAV) and those with bicuspid aortic valves (BAV). At the same time, we examined oxidative stress and the occurrence of SASP in the same samples. Indeed, we report here for the first time that there is an association between increased oxidative damage to telomeres and SASP activation in diseased aortic tissue. Moreover, the activation of SASP leads to the induction of CD163^+^ macrophages and possibly activation of aortic telocytes.

## 2. Results

### 2.1. Patient Characteristics

Healthy thoracic aortic (HTA) and TAA-samples were obtained from 41 and 174 patients, respectively. Subjects in both groups were age- and gender-matched and underwent heart transplantation or surgery that included aneurysm surgery of the TAA. For further analysis, we divided the TAA patients into two groups according to their aortic valve morphology (TAV, tricuspid aortic valve; BAV, bicuspid aortic valve). The number of patients in the TAV group was 109, and in the BAV group, 65. The baseline characteristics of these patients are shown in Table 1. Consistently with the literature [33], there were significant differences between HTA and aneurysmal aortic samples (TAV, BAV) in body mass index (BMI), hypertension, chronic renal failure, and chronic cerebrovascular and coronary diseases (Table 1).

### 2.2. Shortened Telomeres and Telomere Damage in Aneurysm Compared to Healthy Aorta

To determine whether and to what extent telomeres are shortened in aneurysms compared to healthy tissue and whether this has an impact on telomere stability, both TL and DNA damage to telomeres were measured.

Measurement of TL in healthy and diseased aortic tissue showed a significant shortening of telomeres measured by relative qPCR method and quantitative Southern blot method (Figure 1A–D). The relative TL of 174 TAA samples was calculated by qPCR measurement (Figure 1) compared with 41 HTA samples (aortic size < 35 mm). Both TAA groups, TAV and BAV, showed significantly shorter telomeres. When TAA was grouped by effective aneurysm sizes (aortic size 45–55 mm [*n* = 79], 55–65 mm [*n* = 65], and >65 mm [*n* = 30]), the TAA-TAV groups showed significant telomere shortening in all groups, whereas in the TAA-BAV, only the largest group (>65 mm [*n* = 14]) had significantly shorter telomeres (*p* < 0.001; Figure 1A). In the next step, the exact lengths of telomeres were confirmed and determined by Southern blot (Figure 1B–D). This detailed visualization of the exact TL confirmed the results of qPCR and showed significant shortening of telomeres in the TAA-TAV group, and in the TAA-BAV group, which had aortic diameters >65 mm (Figure 1D) only. In human cells, telomerase protein is responsible for telomere maintenance and stabilization of mitochondrial cell homeostasis [34]. Detection of relative telomerase activity (TA), according to previous literature, showed a reduction of TA in TAA, independent of TAV and BAV, compared to HTA (*p* < 0.01; Figure 1E).

Telomere length is a critical factor for direct DNA damage to telomeres [11,29]. Therefore, we measured telomere damage by telomere/γH2AX colocalization (telomere-induced foci, TIFs) (Figure 1F–G). Interestingly, we found a significant increase in TIFs independent of telomere length and valve entity (TAA-TAV vs. TAA-BAV groups). Statistical calculation of TIFs between the different entities of aortic valves showed a slight but nonsignificant difference (data not shown). The increase in telomere-specific H2AX signals compared with the increase in total DNA H2AX signals indicated higher total DNA damage, but again, there was no difference between the subgroups of aneurysm samples and healthy aortic samples (Figure 1G).

In summary, increased telomere damage and shortened telomeres were observed in TAV compared with BAV and healthy aortic tissue. Detailed analysis of the individual samples showed significant shortening of telomeres mainly in the TAV-TAA specimens. In the BAV-TAA group, significant telomere shortening was observed only in aneurysms with a diameter of >65 mm. DNA damage to telomeres was significantly higher in both TAA groups, TAV and BAV, compared with healthy aortas.

### 2.3. GLUT1 and p-eNOS Expression after Hypoxic Stress

The significant increase in DNA damage at the telomeres and in the global DNA seems to be crucial for the shortening of telomeres in the TAA-TAV group. Next, we analyzed the impact of chronic hypoxia in the different aneurysmatic tissues. First, we examined the potential effects of the fixation protocol in the laboratory on DNA damage signaling in the aortic tissues. We fixed and probed a selection of specimens at different time points (Appendix A). Thereafter, only aortic samples processed and fixed in the laboratory within 120 min after collection during surgery were further analyzed. ROS and reactive nitrogen species (RNS) are responsible for the deleterious effects of oxidative stress, and their role in vascular diseases has been well established [35]. OxySelectTM measurement of ROS/RNS indicated a significant increase in hypoxic-induced ROS/RNS production in the TAA group, whereas the HTA group had significantly lower ROS/RNS levels (Figure 2A). Whereas the aortic tissues of TAA-BAV showed a low level of ROS production, the TAV samples showed more pronounced ROS/RNS occurrence.

To provide evidence of chronic hypoxia, we used immunological detection of the glucose transporter GLUT1 (Figure 2B–E). We found that GLUT1 was abundant in the aortic media of aneurysm specimens. In contrast, qualitative inspection of healthy aortic media showed lower GLUT1 expression. Quantification of GLUT1 expression in protein lysates of all aortic specimens confirmed increased expression in TAA-TAV samples compared to the HTA samples (*p* < 0.05, Figure 2D,E). Similarly, TAA-BAV specimens showed slightly increased GLUT1 protein expression, but not significantly, compared to HTA and TAA-TAV specimens (*p* = 0.19). Next, the expression of activated (phosphorylated) eNOS (*p*-eNOS), which affects endothelial function [36,37], was analyzed in correlation with ROS/RNS levels. Western blot analyses demonstrated significantly higher expression of *p*-eNOS in the TAA-TAV group but not in the HTA or the TAA-BAV groups (Figure 2D). Overall, TAA-TAV specimens appeared to trigger GLUT1 and *p*-eNOS expression after hypoxic stress conditions, whereas hypoxic cell damage was lower in HTA and TAA-BAV specimens.

### 2.4. Chronic DNA Damage and Cell Senescence in TAA Specimens

Reaching the critical telomere length, caused either by initially shorter telomeres or by rapid ageing due to exogenous triggers (chronic hypoxia), inevitably leads to direct DNA damage to telomeres and loss of cellular homeostasis. γH2AX, a hallmark of DDR, also forms early during apoptosis and cell senescence [38]. Apoptotic γH2AX staining usually follows a progression that can be divided into three successive patterns: (1) the γH2AX ring, which appears in the early phase of apoptosis as a sign for massive alteration of nuclear size; (2) a pan-staining of the nucleus that maintains its overall morphology and size; (3) the persistence of the pan-staining within apoptotic bodies.

In the first step, we determined cell apoptosis and cell senescence by immunofluorescence staining. Figure 3A,B show the localization of γH2AX in aortic and aneurysmatic specimens. There was a significant increase in γH2AX foci in both TAA groups, TAV and BAV; *p* < 0.01 (Figure 3B). In addition, there was a significant increase in apoptotic rings and full γH2AX staining in both TAA groups (BAV *p* < 0.01; TAV group, *p* < 0.05)

The loss of telomeric stability leads to activation of ATM kinases, which initiate DDR and result in activation of p53 [12]. Therefore, we measured ATM/53BP1 by colocalization immunofluorescence staining (Figure 3C,D). Statistical calculation of ATM/53BP1 pathway activation compared with HTA showed a significant increase in TAA samples compared with HTA control, but no significant difference in TAV and BAV subsets of aneurysm samples (Figure 3D). Micronuclei are a consequence of chronic DNA damage and chromosomal instability [39]. Accordingly, we observed an increase in micronuclei in the TAA specimens compared with the HTA control group (Figure 3E,F). Interestingly, the frequency of micronuclei appeared to be higher in the BAV subgroup, but not significantly (Figure 3F, and data not shown). These observations confirmed increases in chronic DNA damage, cell senescence and chromosomal integrity in specimens obtained from TAA-TAV and TAA-BAV patients.

### 2.5. Expression of Senescence-Associated Secretory Phenotype (SASP) in Aneurysmatic Aortic Tissue

As previously described, critical lengths of telomeres lead to direct telomere DNA damage and impair cellular function and metabolism. This leads to senescence and changes the phenotype of the cell to a senescence secretory phenotype [40]. Therefore, our next interest was to determine whether the observed γH2AX-related senescent cells express SASP secretomes.

First, we measured the SASP-related chemokine IL-6 and phosphorylated (*p*)-Rb protein expression in aortic specimens (Figure 4A,B). IL-6 and *p*-Rb proteins’ expression was significantly increased in TAA-TAV samples relative to GAPDH expression (*p* < 0.01 and *p* < 0.05, respectively), which correlated with a significant increase in the cell cycle regulator p21 (*p* < 0.01), which plays a crucial role in SASP activation (Figure 4A,B). In contrast, significant expression of chemokines and regulatory proteins was absent in the HTA and aneurysm subgroup TAA-BAV. These factors are reportedly induced by multiple mechanisms, including nuclear factor-kB (NF-κB) [41] and mammalian target of rapamycin (mTOR) signaling [42] in senescent cells. Measurement of mTOR and NF-κB protein expression was performed by IF-staining (Figure 4C,D). Both proteins were significantly increased in TAA-TAV (*p* < 0.01) and TAA-BAV (*p* < 0.05) compared to HTA specimens; the TAV group tended to have increased levels. The higher number of senescent cells in tissue may be due to increased infiltration of the senescent immune cells. In this context, CD163^+^ macrophages were shown to be frequently susceptible to senescence [43,44]. IF staining of aneurysmatic tissue showed an increase in CD163-positive immune cells (Figure 4E,F). Similar to the results of cytokine and cell cycle regulatory protein expression shown above, there is a clear trend that TAA-TAV patients have significantly more SASP-involved CD163^+^ macrophages (*p* < 0.01), compared to HTA and the aneurysmatic subgroup TAA-BAV (*p* < 0.05).

### 2.6. Chronic Hypoxia and Cellular Stress Correlates with the Number of Telocytes

Recently, we showed that telocytes are involved in HTA, and the number of telocytes was positively correlated with increases in chronic disease and aortic diameter in TAA samples [45,46]. An increase in the number of TCs seems to be positively related to chronic hypoxia and/or cellular stress of the tissue. Therefore, in the next experiments, we wanted to investigate the relationship among chronic damage to telomeres, senescence, and the number of telocytes in diseased aortic tissue.

We used the colocalization of the markers CD34, ckit, and PDGFR-β in IF staining to detect telocytes (Figure 5A,B). We analyzed the presence of telocytes i) in the whole tissue and ii) selectively in intimal layer of the aorta (Figure 5C). Quantitative analysis showed significant increases in the number of telocytes in the intimal layer (*p* < 0.001) and the whole tissue (*p* < 0.01) and the appearance of γH2AX (*p* < 0.0001) and TIFs (*p* < 0.001) in TAA, compared with HTA (Figure 5C). In addition, there is a clear trend that the TAA-BAV samples had more telocytes than the aneurysmal subset TAA-TAV (Figure 1D). There were significant positive correlations between the increase in telocytes found in the tissue and the number of TIFs (R = 0.5433 *p* < 0.001, Figure 5E) and the total γH2AX signals (R = 0.6392 *p* < 0.001, Figure 5F). Concomitant detection of telomeric DNA damage and global DNA damage in telocytes could not be detected (data not shown). In summary, a relationship among shortened telomeres, DNA damage, and the increase in telocyte number can be demonstrated.

## 3. Discussion

While several causes, such as chronic hypoxia and tissue remodeling, are relatively well understood regarding the development of aneurysms, the role of telomere degradation is not fully understood. Telomere shortening, or structural damage may be the result of ROS, which further leads to the expression of SASP and degenerative tissue remodeling, especially in organs and tissues with low regenerative potential [27,47]. The aim of this study was to investigate the role of telomeric DNA damage caused by cellular oxidative stress, which promotes distributed cell division and death. The activation of SASP in these cells leads to the expression of factors involved in extracellular matrix (ECM) destruction. This suggests the presence of a replication-independent mechanism of telomere shortening in the development of aortic aneurysm. This suggestion is supported by the following observations: (i) Telomeres are known to be shortened in the thoracic aorta, but the significance decreases in genetically predisposed tissue weaknesses as in the aortic valve etiology of BAV [1,7,48,49,50]. (ii) The TAA-BAV group exhibits less cellular hypoxic damage (lower ROS/RNS and GLUT1 levels) but appears to be as fragile as the TAA-TAV group due to similar incidence of DDR of telomeric DNA. (iii) The inhibition of DNA repair caused by hypoxic conditions could lead to genetic rearrangement and genomic instability [39]. (iv) Telomerase is known to play a non-canonical role in suppressing ROS generation, which contributes to the maintenance of mitochondrial function and prevents SASP expression [51,52]. Low TA levels, as observed in TAA, would enhance cellular senescence and ECM degradation. v) Senescence secretory phenotype activates immune cells and leads to chronic systemic inflammation [40]. This chronic, low-grade systemic inflammation is the result of a process called inflamm-aging, a crucial pathogenic mechanism in multiple age-related diseases [53].

The observation that both ROS and telomere dysfunction are associated with age-related diseases suggests that the biology of ROS and telomeres is coupled with disease causation [54]. This is supported by the hypothesis that many diseases risk factors, such as age, smoking, and stress, increase oxidative stress and telomere deficiency. Oxidative stress causes oxidative damage to telomeric ends and induces telomeric DDR, one of the critical aspects involved in the process of cellular senescence [54]. We compared telomere lengths in aneurysms from patients with risk factors favoring the development of TAA and patients genetically predisposed to it. We observed evidence of increased chronic hypoxic stress, and the eNOS response was completely absent in the TAA-BAV group. This may indicate a less adequate response to ROS in the TAA-BAV group. Further, a significant increase in telomere-induced DNA damage was observed in both TAA-TAV and TAA-BAV samples, regardless of the effective length of telomeres. Interestingly, both groups, TAA-TAV and TAA-BAV, showed the same occurrence of γH2AX-DDR signals at the ends of telomeres (TIFs) and throughout the genome (Figure 1G). These observations in our study are in accordance with those of Balint et al., who recently described similar results using comparable methods. Their analysis of TAA-BAV compared with TAA-TAV specimens showed a significantly higher proportion of senescent aortic cells (p21/β-Gal positive) and γH2AX-positive cells [55].

γH2AX staining for the detection of DDR in aortic samples was divided into three timely sequential patterns: (i) the focal patterns of DNA damage foci caused by DSB, (ii) a ring stain for the detection of the early apoptotic phase with no change in nuclear size, and (iii) pan staining of the nucleus while maintaining its overall morphology and size [56]. In our work, the calculation of apoptotic rings and γH2AX foci was to evaluate early apoptosis versus DNA damage response in our clinical samples. Thus, we found that the TAA-BAV group had the highest occurrence of early apoptotic cells (Figure 3B). Moreover, the apoptotic ring is an intrinsic marker of apoptotic DNA breaks that initiate the recruitment of other DDR signals such as ATM, Chk2, and DNA-PK [56], indicating a chronic event that initiates regulatory DNA repair mechanisms.

However, micronuclei are known biomarkers of DNA damage and sources of pro-inflammatory DNA [57]. They have been reported to accumulate DDR proteins such as p53, H2AX, ATM, Rad71, and SMAD7, the latter being markers for DNA replication stress [58]. Previous studies have shown that DNA leaking from disrupted micronuclei triggers the innate immune mechanism, cGAS-STING, which promotes inflammation that can cause a wide range of age-related diseases [59]. In this work, we observed that micronuclei were significantly increased in TAA compared to HTA specimens. In addition, the TAA-BAV group had a higher number of micronuclei as compared to the TAA-TAC group. As micronuclei trigger the immune system and are part of inflamm-ageing, micronuclei may contribute to immunopathology in these “predisposed” tissues. Ongoing studies are investigating the differences between genetically predisposed patients at high risk of aneurysm (e.g., Marfan disease, Ehlers–Danlos, and BAV etiology) and patients at high chronic risk (smokers, hyperlipidemia, etc.) with respect to micronuclear occurrence and accumulation. Moreover, it should be noted that SASP factors attract immune cells such as macrophages and epithelial progenitor cells (EPCs) [60]. Recently, our group described the presence of telocytes in aortic tissue and a significant increase in these cells with devastating consequences in aneurysmal tissue [46]. In particular, our study characterized the function and location of TC exosomes in TAA and observed the recruitment of progenitor cells from EPCs subsets. This indicates that TCs in the aorta are involved in smooth-muscle-lineage development during aneurysm formation. However, how the attraction of TCs occurs remains to be elucidated. Is chronic hypoxic stress responsible for the attraction of TCs in aneurysmal tissue, or are they only involved in the induction and proliferation of TCs? The correlations of TC and hypoxic DNA damage suggest the second option. Indeed, the significant increase in TCs showing expression of negatively modifiable factors on vSMC has devastating consequences for disease progression. Previous studies have shown that telomerase has two functions in cells: canonical and non-canonical [61]. While the canonical function is associated with telomeres, the non-canonical plays a role in suppressing ROS generation, which is part of mitochondrial maintenance and prevents the SASP phenotype [52]. Telomerase is an enzyme consisting of the catalytic reverse transcriptase (TERT) subunit and non-coding RNA (TERC), which serves as a template for telomere elongation by TERT [62]. Under the influence of oxidative stress, TERT can migrate from the nucleus to the mitochondria, where it contributes to the inhibition of ROS production [63]. Inhibition of TA detected in human SMCs induced senescence and showed reduced NOS activity [64]. In this work, we observed that both aneurysm entities, TAV and BAV, exhibit reduced TA compared with HTA (Figure 1E). As previously described, reduced TERT activity increases mitochondrial and cellular oxidative stress, which promotes distributed cell division and cell death [65].

In summary, compared to TAA-TAV and HTA samples, cells in TAA-BAV samples undergo stable cell cycle arrest, accompanied by morphological and phenotypic changes associated with cellular senescence [66]. This ultimately leads to the upregulation of SASP-related proteins, promotes chronic inflammation, and accelerates age-related functional decline and disease [67]. This calls for drugs targeting cellular senescence. Specially, senolytic agents that induce cell death in senescent cells, IL-1 receptor antagonists, or the use of NF-κB or mTOR inhibitors are likely to open new therapeutic opportunities. Nevertheless, inhibition of SASP is a promising approach for future clinical implementation to promote healthy aging.

## 4. Materials and Methods

### 4.1. Patients’ Specimens

This study was approved by the Ethical Committee of the Medical University of Vienna (EK 1280/2015). Written informed consent was obtained from all patients prior to inclusion in the study. The investigation conformed to the principles that are outlined in the Declaration of Helsinki regarding the use of human tissue. Human aortic tissue samples (215 samples) were obtained either during heart transplantation (41 samples) or during the surgical procedure, which involved aneurysm surgery of TAA (174 samples). Exclusion criteria were patients with ongoing endocarditis, sepsis, recent infectious disease, or genetic disorders; the intake of immunomodulation therapy (e.g., cortisone); anti-tumor therapy. After receiving the specimens, aortic tissue was sliced. One part was snap-frozen and stored in liquid nitrogen, and the other part was fixed in 4.5% formalin.

### 4.2. Telomere Length Measurement by PFGE/Southern Blot

The telomere length was measured by Southern blot, as previously described [68]. Briefly, Southern blot analyses were performed with TeloTAGGG Kit (Roche, Sigma-Aldrich, Vienna, Austria) following the manufacturer’s protocol. The number of samples which were analyzed was limited to the control group’s 16 HTA samples, and the two TAA groups—24 TAV samples and 20 BAV samples. One microgram of genomic DNA was enzymatically digested with RsaI and Hinf1 for 2–15 h. Digested DNA was separated on a 1% agarose gel (Bio-Rad Laboratories, Vienna, Austria; Pulsed-field certified) and electrophoresed for 15–20 h at 30 V. Electrophoretically separated DNA fragments were vacuum-blotted onto a positively-charged nylon membrane (Roche, Sigma-Aldrich, Vienna, Austria) with a VacuGene XL apparatus (GE Healthcare, Vienna, Austria). We applied depurination (0.25 N HCl for 30 min), denaturation (0.5 M NaOH, 1.5 M NaCl, 30 min), neutralization (0.5 M Tris-HCl, 3 M NaCl, pH 7.5, 30 min), and blotting (3 M NaCl, 0.3 M Na-citrate, pH 7, 2 h) steps. DNA was cross-linked with the membrane by UV irradiation (Bio-Rad GS Gene Linker UV Chamber, 50 mJ). Detection of telomere fragments was performed with the Telo TAGGG Telomere length assay (Roche, Sigma-Aldrich, Vienna, Austria) by using an AP-conjugated anti-DIG antibody and visualized by X-ray, according to the manufacturer’s instructions. Image quantification was performed using Image Quant software to measure the intensity value of each telomere smear. The intensity value of each sample was then matched to the background value of a lane without a DNA sample.

### 4.3. Relative Telomeric Length by RT-qPCR

The average telomere length ratio was measured from total genomic DNA as previously described [69]. Briefly, The average telomere length ratio was measured from total genomic DNA using a real-time PCR assay [69]. DNA was extracted with DNeasy kit (Qiagen, Hilden, Germany) and treated with DNase I free RNase A (Ambion, Kaufungen, Germany). cDNA was used in Maxima^®^ SYBR Green/ROX qPCR Master Mix (2×) (Fermentas Thermo Scientific™, Glen Burnie, MD, USA) with the 36B4 single-copy gene and a gene-specific probe in a 7500 Real-Time PCR system (Applied Biosystems–Fisher Scientific, Hagen, Germany). Samples were normalized to single-copy genes as indicated, and fold change was determined by ΔΔCt method [70]. The primer sequences for qPCR were TEL-1-CGGTTTGTTTGGGTTTGGGTTTGGGTTTGGGTTTGGGTT, TEL-2-GGCTTGCCTTACCCTTACCCTTACCCTTACCCTTACCCT, 36B4u-CAGCAAGTGGGAAGGTGTAATCC, and 36B4d-CCCATTCTATCATCAACGGGTACAA).

### 4.4. Telomerase Activity Assay

For quantitative analysis of TA, a Telomeric Repeat Amplification Protocol (TRAP) and a photometric enzyme immunoassay were performed in all samples using a TeloTAGGG Telomerase PCR ELISA kit (Roche Diagnostics, Vienna, Austria), according to the manufacture’s protocol [71]. The numbers of samples were, for the HTA group, 41; for TAV, 109; and for BAV, 65.

### 4.5. Immunofluorescence Staining and Microscopy

For immunohistological staining, all aortic tissue samples (*n =* 215) were fixed in 4% PBS-buffered formaldehyde. The tissues were embedded in paraffin, deparaffinized with HistoSAV, and rehydrated in a descending series of ethanol. Following heat-induced antigen retrieval with citrate buffer (pH 6), the sections were blocked (10% goat serum, 1% BSA, and 0.1% Tween-20 in PBS) at RT for 60 min. Samples were incubated with 2–5 µg of primary antibody O/N, followed by incubation with an appropriate secondary antibody, and mounted in Vectashield mounting medium including 0.5 µg/mL DAPI (Vectashield; VectorLabs, Burlingame, CA, USA). Negative controls were obtained following the same protocol, but omitting the primary antibodies, and with the usage of purified anti-mouse and anti-rabbit IgG (Abcam, Cambridge, UK). For confocal microscopy, we used a LSM700 Meta microscopy laser system, the appropriate filters, and a ZEN 2010 microscopy system (Zeiss, Inc. Jena, Germany). For statistical analysis of γH2AX, TIF, and micronuclei staining, all samples were analyzed, whereby the mean counted and analyzed cells per group were for the HTA group, 5707 analyzed cells (mean 139.2 cells/sample), the TAV group, 19,421 analyzed cells (mean 178.2 cells/sample), and the BAV group, 13,530 analyzed cells (mean 208.2 cells/sample). For ATM/p53 staining, the numbers of counted cells were for HTA, 3922 cells; for TAV samples, 11,568 cells; and for BAV, 9437 cells.

### 4.6. Western blot Analysis and ROS/RNS Measurements by OxySelect Assay

Aortic samples (equal number of each group, HTA *n* = 13; TAV, *n* = 13; BAV, *n* = 13) were dissected, minced, and dissolved in 250 μL of ice-cold modified RIPA buffer (50 mM Tris-Cl (pH 7.4), 150 mM NaCl, 1 mM EDTA, 1% NP-40, 0.25% Na-deoxycholate, 1 mM PMSF, 10 μg/mL aprotinin, 10 μg/mL leupeptin, 1 mM Na_3_VO_4_, and 1 mM NaF) [72]. The lysate was rotated 360° at 4 °C for 1 h, followed by centrifugation at 12,000× *g* at 4 °C for 10 min. Total protein was quantified using the Bradford protein assay kit (Biorad), and equal amounts of protein were resolved on an SDS-polyacrylamide gel and transferred to a nitrocellulose membrane. Immunodetection was performed by blocking the membranes for 1 h in TBS buffer (20 mM Tris-Cl (pH 7.5), 137 mM NaCl, 0.05% Tween-20) containing 5% powdered non-fat milk, followed by addition of the primary antibody in TBS buffer for 2 h at RT. Primary antibodies were detected with peroxidase-coupled secondary antibodies and developed by enhanced chemiluminescence (ECL system, Amersham Pharmacia Biotech Inc., Arlington Heights, IL, USA) according to manufactures’ instructions. For measurement of ROS in aortic tissue. Superoxide levels in aortic tissues were measured in triplicate using the OxiSelect^TM^ In Vitro ROS/RNS Assay Kit according to the manufacturer’s protocol (Cell Biolabs, Inc., San Diego, CA, USA). A H_2_O_2_ standard curve was prepared for each plate and statistical analysis as previously described.

### 4.7. Antibodies

Primary antibodies were γ-H2AX (Abcam, Cambridge, UK, Anti-gamma H2A.X (S139 antibody [9F3], ab26350) or GAPDH (14C10) rabbit IgG mAb (Cell Signaling Technology, Cambridge, UK CST, 2118), anti-ckit (Abcam, ab32363), Anti-eNOS (Cell Signaling Technology, 9572), or PDGFR-b (Abcam, ab69506); anti-beta Actin (Abcam, ab8227), anti-IL6 (Abcam, ab6672), anti-phospho-Rb (Abcam, ab47763), anti-p21 (Abcam, ab109520), or anti-CD163 (Abcam, ab156769); anti-mTOR (Abcam, ab2732), anti-NFkB (Santa Cruz Biotechnologies, Inc., Dallas, TX, USA, sc8008), anti-ATM (Abcam, ab32420), or anti-GLUT1 (Abcam, ab115730); CD34 (Santa Cruz Biotechnologies, Inc., Dallas, TX, USA, sc74499). Secondary antibodies: AF488GAM (Molecular Probes, ThermoFisher Scientific, Waltham, MA, USA; Novus Biogicals, LLC, Centennial, CO, USA, A11029), AF546GAR Molecular Probes, A11035), HRP-anti-mouse IgG (BD Pharmigen^TM^ (BD Biosciences), San Jose, CA, USA, 7076S), and HRP-anti-rabbit IgG (BD Pharmigen^TM^, 7074S). 

### 4.8. Statistical Analysis

Data from independent experiments were expressed as mean ± standard deviation (SD). Statistical analysis was performed with GraphPad Prism Version 6 (GraphPad Software, San Diego, CA, USA) and SPSS 15.0 software (SPSS Inc, Chicago, IL, USA). Comparisons between the groups were performed by two-tailed Student’s *t*-test or analysis of variance (ANOVA). For comparisons between multiple groups, Kruskal–Wallis tests followed by Bonferroni correction were applied. The correlations between two continuous variables were assessed with Pearson’s test or a non-parametrical Spearman correlation test. *p* < 0.05 was considered statistically significant.

## Figures and Tables

**Figure 1 ijms-23-15498-f001:**
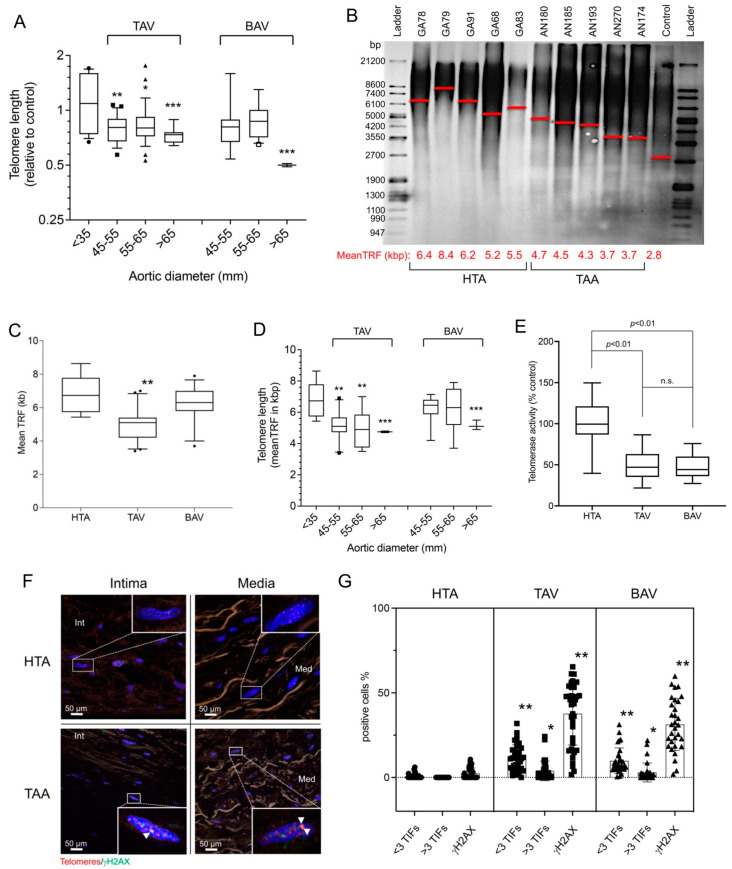
Increase in telomere damage and shortened telomeres in thoracic aortic aneurysm. Measurement of telomere length (TL) in healthy and diseased aortic tissues showed a significant shortening of telomeres and an increase in telomere-specific DNA damage. (**A**–**D**) Measurement of telomeres was performed by relative qPCR method and quantitative Southern blot method. (**A**) Statistical analysis of qPCR measurement from the two groups TAV (tricuspid aortic valve (*n* = 109)) and BAV (bicuspid aortic valve, (*n* = 65)) compared to healthy aortic specimens (<35 mm, (*n* = 41)). The x-axis shows the individual sizes of the aortic tissues at the time of surgery. Specific analysis of TL in kilobase pairs (kbp) was carried out using Southern blot analysis (HTA samples, *n* = 16, TAA samples, *n* = 44). (**B**) shows a representative blot of the measurements. HTA, human thoracic aorta; TAA, thoracic aortic aneurysm. Mean TRF is shown in kbp along the x-axis and marked as a red line in the blot. Molecular ladder in base pairs (bp) are shown to the left and right of the blot. (**C**) Statistical analysis of the telomeric Southern blot evaluation for the control group HTA (*n =* 16) and the two TAA groups, TAV (*n =* 24) and BAV (*n =* 20), separately. Subfigure (**D**) shows the statistical analysis of the quantitative TL divided into the three groups HTA, TAV, and BAV, and along the x-axis divided according to the size of the aorta. (**E**) Relative telomerase activity measured in isolated proteins of all human samples by telomerase TRAP-ELISA assay (HTA group (*n =* 41) and the two TAA groups, TAV (*n =* 109) and BAV (*n =* 65)). n.s., non-significant. The measurement of telomere damage was performed by immunohistochemistry. (**F**) Representative images of the staining (red, telomeres; green, γH2AX) in the different aortic layers (intima and media). Cell nuclei were stained by DAPI (blue). Co-localizations, telomere-induced foci (TIFs), occurred more frequently in the TAA group vs. the HTA group. (**G**) Statistical analysis of TIFs divided into <3 TIFs, >3 TIFs, and γH2AX alone, plotted separately for 41 HTA samples (5707 analyzed cells), TAV samples (19,421 analyzed cells), and 65 BAV samples (13,530 analyzed cells). * *p* < 0.05; ** *p* < 0.01; *** *p* < 0.001.

**Figure 2 ijms-23-15498-f002:**
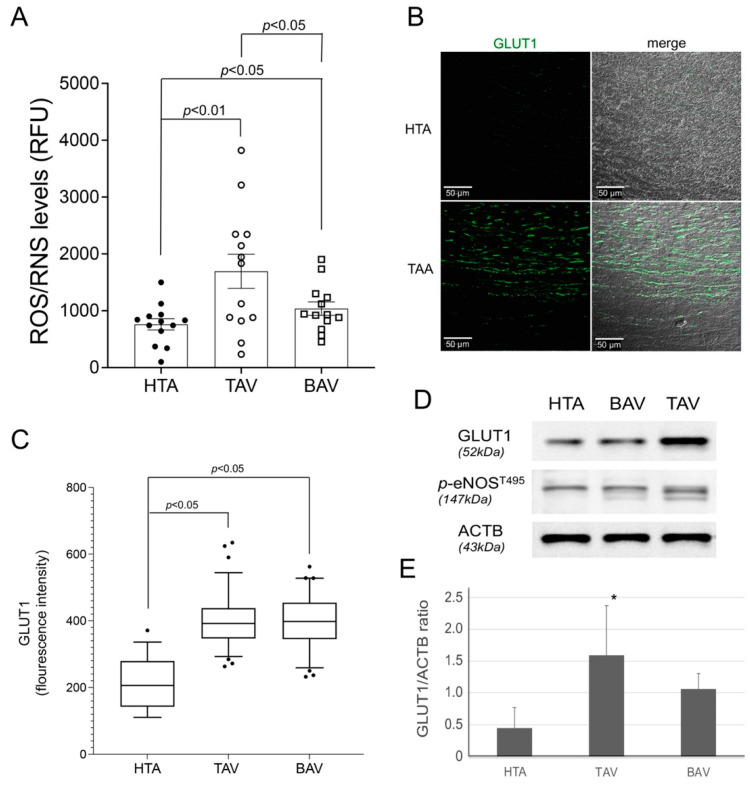
ROS/RNS and GLUT1 expression was detected in aneurysm specimens. Measurement of ROS/RNS and GLUT1 protein levels indicated activation during hypoxic stress in aneurysm sections. (**A**) ROS/RNS measurement by OxiSelect™ in protein extracts of healthy aortic (HTA, *n* = 13) and aneurysm (TAV, *n* = 13, BAV, *n* = 13) specimens. RFU, relative fluorescence units. GLUT1 expression was measured in all aortic samples by immunofluorescence (IF, (**B**,**C**)) and confirmed in selected samples (13 samples of each group) by Western blot (**D**,**E**). (**B**) Representative image of GLUT1 protein (green) expression stained by IF. Scale bar, 50 µm. (**C**) Statistical analysis for GLUT1 protein expression in HTA (4092 cells), compared to TAA-TAV (18,293 cells) and TAA-BAV (11,543 cells) specimens. (**D**) The blot shows representative GLUT1 and *p*-eNOS specific results, and the reference beta-actin (ACTB). (**E**) Statistical analysis for GLUT1 protein expression detected by Western blot is given (HTA, *n* = 13; TAV, *n* = 13; BAV, *n* = 13). * *p* < 0.05.

**Figure 3 ijms-23-15498-f003:**
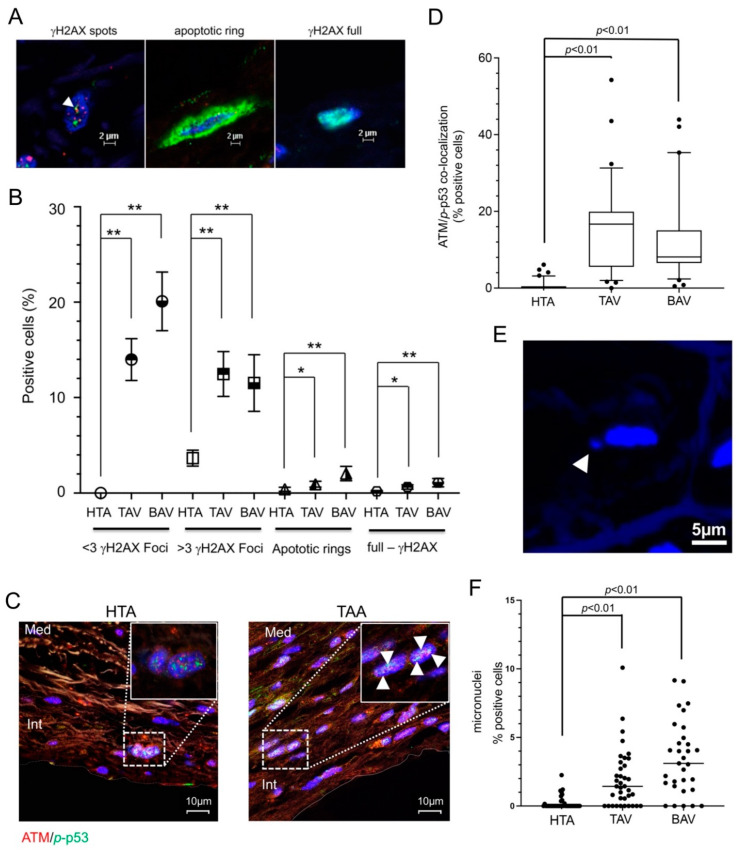
DNA damage, cell senescence and cell death measured by γH2AX/ATM/*p*-p53 staining. Hypoxic stress in aneurysm sections leads to DNA damage and cellular senescence measured by γH2AX IF-staining. (**A**) Representative images of γH2AX spots (white arrow, left image), apoptotic ring (middle image) and γH2AX full staining (right image) detected in aortic and aneurysmal specimens. Green, γH2AX; blue, cell nuclei (DAPI). Scale bar, 2 µm. (**B**) Statistical calculations of different amounts of γH2AX foci (<3 foci and >3 foci), apoptotic rings, and full-γH2AX staining measured in HTA (*n* = 41) and both TAA groups, TAV (*n* = 109) and BAV (*n* = 65). *n* of counted cells: HTA, 5707; TAV, 19,421 cells; and BAV, 13,530 cells. * *p* < 0.05; ** *p* < 0.01. ATM and *phosphorylated (p)*-p53 expression was measured by IF staining (**C**,**D**). (**C**) Representative images of ATM protein (red) and *p*-p53 (green) expression stained by IF. Colocalizations were found in TAA samples (white arrows). Blue, cell nuclei (DAPI). Med, media; Int, intima. Scale bar, 10 µm. (**D**) Statistical analysis for ATM and *p*-p53 protein colocalization in HTA, compared to TAA-TAV and TAA-BAV specimens. ATM/p53 staining measured in HTA samples (*n* = 23) and both TAA groups, TAV samples (*n* = 61) and BAV samples (*n* = 53). Numbers of counted cells of each group: HTA, 3922 cells; TAV, 11,568 cells; and BAV, 9437 cells. (**E**) The image shows representative micronuclei next to cell nuclei stained by DAPI (blue). (**F**) Statistical analysis for positive cells with micronuclei (%) detected by IF is given.

**Figure 4 ijms-23-15498-f004:**
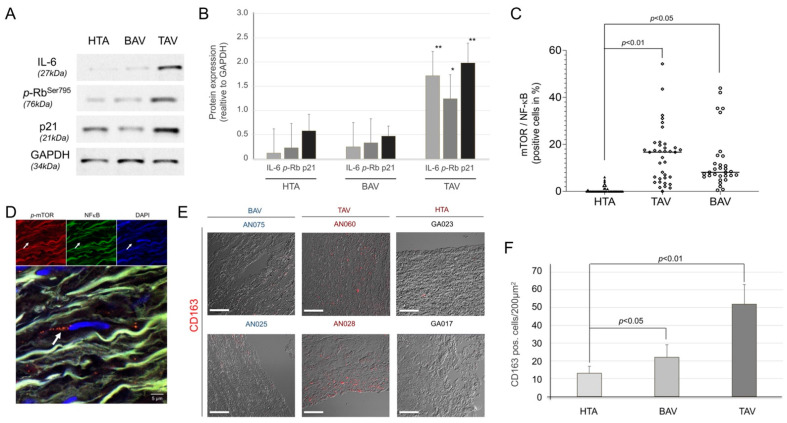
Senescence and SASP expression in healthy and diseased thoracic aortic tissue. Expression of cellular marker, a chemokine (IL-6), cell cycle regulatory proteins (Rb, p21), and SASP regulatory proteins (mTOR and NF-κB). (**A**) Representative blot of IL-6, p-Rb, and p21 measured by Western blot, GAPDH was used as an internal control. Protein size is given in kilodaltons (kDa). HTA, human thoracic aorta; TAV, tricuspid aortic valve; BAV, bicuspid aortic valve. (**B**) Statistical analysis for protein expression detected by Western blot in HTA (*n* = 13), compared to TAA-TAV (*n* = 13) and TAA-BAV (*n* = 13) specimens. Protein expression was calculated relative to internal control (GAPDH). * *p* < 0.05; ** *p* < 0.01. NF-κB/*p*-mTOR protein expression given as colocalization (white arrow) was detected by IF-staining. (**C**) Statistical calculations for HTA (*n* = 24), TAV (*n* = 37), and BAV (*n* = 30)). A representative image of IF staining is shown (**D**). Red, *p*-mTOR; green, NF-κB; blue, cell nuclei (DAPI). (**E**) Representative image of CD163^+^ macrophages detected in aortic and aneurysmal specimens. Scale bar, 100 µm. (**F**) Statistical calculations of CD163-positive cells/200 µm^2^ in HTA, TAV, and BAV specimens.

**Figure 5 ijms-23-15498-f005:**
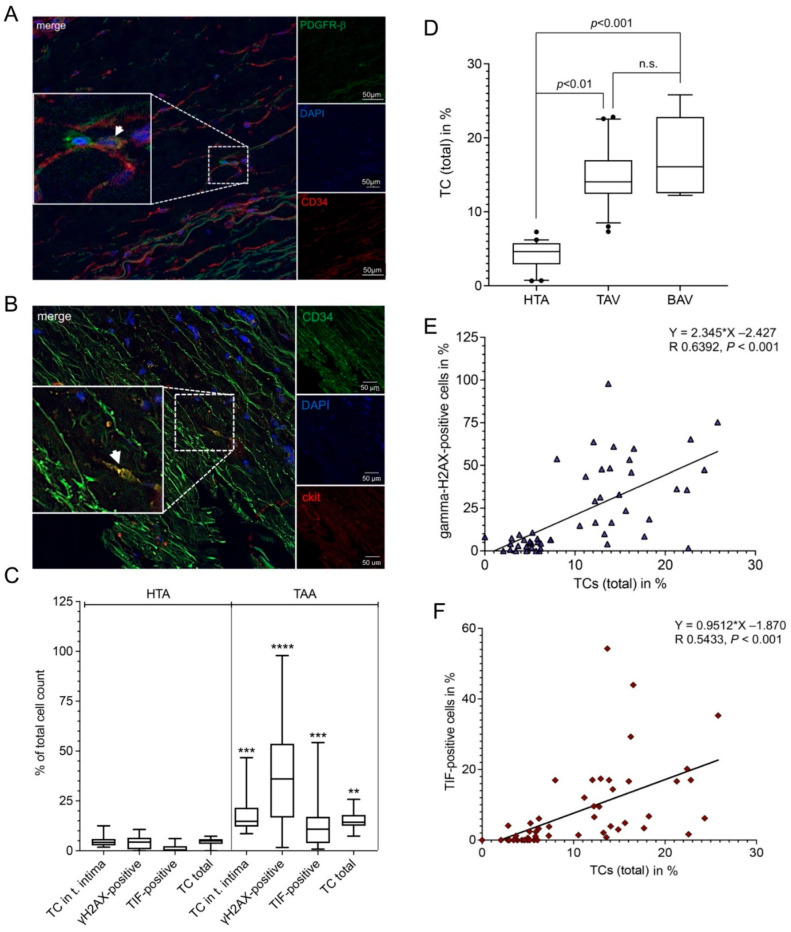
The amounts of DNA damage and telomere-induced foci correlate with the number of telocytes (TCs). The number of TCs (CD34^+^/ckit^+^) and the amounts of telomeric (γH2AX/telomere co-localization) and global DNA damage (>3 γH2AX foci) were measured by immunofluorescence. (**A**) Representative image from the detection of TCs. TCs were characterized as (**A**) CD34 (red) and PDGFR-β (green) double positive, or (**B**) ckit (red) and CD34 (green) double positive staining. Double positive TC is shown by the white arrow. Scale bar, 50 µm. Blue, cell nuclei (DAPI). Scale bar, 2 µm. Statistical analysis (**C**) shows significant increases in TC, γH2AX-positive (>3 γH2AX foci), telomere-induced foci (TIF)-positive, and total TCs found in thoracic aortic aneurysm (TAA) compared to healthy thoracic aortic (HTA) specimens. ** *p* < 0.01; ***, *p* < 0.001; ****, *p* < 0.0001. (**D**) Statistical calculations of the percentage of TCs in HTA, TAV, and BAV specimens. n.s., non-significant. Correlation of TCs with (**E**) γH2AX-positive signals and (**F**) TIF-positive cells found in aortic tissue samples showed significant correlation in both calculations (% of TC and γH2AX^positive^cells, *p* < 0.001; % of TC and TIF^-positive^ cells, *p* < 0.001). Total cell count per group, HTA = 4767 cells; TAA = 17,954 cells.

**Table 1 ijms-23-15498-t001:** Patient baseline characteristics.

	Study Population		HTA		TAV		*p* Value	BAV		*p* Value
	(*n* = 215)		(*n* = 41)		(*n* = 109)			(*n* = 65)		
*Demographic, risk factors, and comorbidities*										
Age (years) (range)	60.1	(19–81)	54.9	(20–79)	62.9	(23–81)	<0.01	58.3	(19–80)	0.12
female, *n* (%)	73	(33.9)	8	(19.5)	51	(46.8)	<0.05	14	(21.5)	0.50
Body mass index (BMI), *n* (range)	27.2	(15–45)	25.3	(19–30)	26.7	(15–45)	<0.05	29.0	(18–44)	<0.01
Adipositas (BMI >30), *n* (%)	57	(26.5)	2	(4.8)	27	(24.8)	<0.05	28	(43.1)	0.18
Smoker, *n* (%)	58	(27.0)	18	(43.9)	20	(18.6)	0.39	20	(30.7)	0.10
Hypertension, *n* (%)	149	(69.3)	18	(43.9)	85	(78.0)	<0.01	46	(70.8)	<0.01
Dyslipidaemia, *n* (%)	93	(43.6)	17	(41.5)	46	(42.2)	0.45	30	(46.2)	0.27
Chronic renal failure, *n* (%)	31	(14.4)	15	(36.6)	11	(10.1)	<0.01	5	(7.7)	<0.01
Diabetes, *n* (%)	25	(11.6)	8	(19.5)	9	(8.3)	0.18	8	(12.3)	0.05
COPD, *n* (%)	35	(16.3)	3	(7.3)	25	(23.0)	<0.05	7	(10.8)	0.31
Positive family history, *n* (%)	8	(3.7)	1	(2.4)	3	(2.8)	0.46	4	(6.2)	0.17
CAD, n (%)	68	(31.6)	20	(48.8)	32	(29.4)	<0.05	16	(24.6)	<0.05
CVD, *n* (%)	18	(8.4)	7	(17.1)	3	(2.8)	<0.01	8	(12.3)	<0.01
PAD, *n* (%)	19	(8.8)	4	(9.8)	7	(6.4)	0.26	8	(12.3)	0.18
*Therapeutics*										
Oral diabetes therapy, *n* (%)	21	(9.8)	7	(17.1)	7	(6.4)	0.06	7	(10.8)	0.18
Statins, *n* (%)	68	(31.6)	14	(34.1)	34	(31.2)	0.38	20	(30.8)	0.38
Aspirin, *n* (%)	63	(29.3)	17	(41.5)	26	(23.9)	**<0.05**	20	(30.8)	0.15
b-Blocker, *n* (%)	98	(45.6)	20	(48.9)	52	(47.8)	0.45	26	(40.0)	0.21
ACE-Inhibitor, *n* (%)	80	(37.2)	19	(46.3)	39	(35.8)	0.13	22	(33.9)	0.12

COPD, chronic obstructive pulmonary disease; CAD, coronary artery disease; CVD, cerebrovascular disease; PAD, peripheral arterial disease.

## Data Availability

The data that support the findings of this study are available from the corresponding author upon reasonable request.

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
