# Peer review of "Impacts of Telomeric Length, Chronic Hypoxia, Senescence, and Senescence-Associated Secretory Phenotype on the Development of Thoracic Aortic Aneurysm"

_ijms, 2022, doi:10.3390/ijms232415498_

Round 1

Reviewer 1 Report

It is interesting that the authors tested several parameters or characteristics that might impair the development of TAA. And it would be helpful for the following researchers if the authors could improve the quality of the images, label the accurate numbers of tested samples for statistical analysis, and describe the methods part more detailed. Some of the points were listed and hope they will help to improve the manuscript finally. 

1.     In Fig. 1, the authors presented the length of telomeres of HTA and TTA. In 1A, relative qPCR results were shown and relative to HTA as described, but the numbers of samples of each group were not labelled. In 1B, the TLs were not accurate in my eyes. TL calculation should involve parameters of both length and intensity, so the authors should present how they calculate the TLs in supplementary. Besides, only a small part of the samples was shown by southern blot. What about the others? In the rest of the figures, the numbers of samples of each group should also be labelled. 

2.     Supplementary Fig. 1 A+B were mentioned in line 122 to indicate the TL, but the S.-Figure 1 displayed the DNA damage data, not the TL data.

3.     In Fig. 1F, typical telomere FISH-IF was performed using the tissue samples, but the detailed methods were not described. Besides, the signals of both telomeres and pH2AX were not clear enough. Much higher resolution pictures and bigger typical views should be presented for readers. 

4.     In Fig.3 B, D and F, how many cells were counted for statistical analysis? 

5.     In Fig.4E, the resolution of the images was too low to see them clear. 

Reviewer 2 Report

congratulations for the work, from my point of view you have achieved a very complete article, where they study the relationship of different factors with aortic aneurysm. The central theme is the shortening of telomeres and its association with disease. In the article there is also a cellular relationship to the disease, which it demonstrates with clear and concrete details. A relationship between macrophages and the appearance of the disease is observed. I consider that it is an article that should be published. It supposes a clear vision at a scientific level of a cause-effect association with the shortening of telomeres and disease. I consider that if we add the scientific quality with the correct exposition in the work, it should be published and it will be interesting for potential readers

Author Response

Comments and Suggestions for Authors

congratulations for the work, from my point of view you have achieved a very complete article, where they study the relationship of different factors with aortic aneurysm. The central theme is the shortening of telomeres and its association with disease. In the article there is also a cellular relationship to the disease, which it demonstrates with clear and concrete details. A relationship between macrophages and the appearance of the disease is observed. I consider that it is an article that should be published. It supposes a clear vision at a scientific level of a cause-effect association with the shortening of telomeres and disease. I consider that if we add the scientific quality with the correct exposition in the work, it should be published and it will be interesting for potential readers

Reply: Thank you very much! We are very pleased to read such a positive review.

Round 2

Reviewer 1 Report

Glad to see the improved quality of the manuscript. I don't have more comments.